# A Review of Non-Destructive Testing (NDT) Techniques for Defect Detection: Application to Fusion Welding and Future Wire Arc Additive Manufacturing Processes

**DOI:** 10.3390/ma15103697

**Published:** 2022-05-21

**Authors:** Masoud Shaloo, Martin Schnall, Thomas Klein, Norbert Huber, Bernhard Reitinger

**Affiliations:** 1LKR Light Metals Technologies Ranshofen, Austrian Institute of Technology, Lamprechtshausenerstraße 61, 5282 Ranshofen, Austria; martin.schnall@ait.ac.at; 2RECENDT Research Center for Non Destructive Testing GmbH, Science Park 2/2. OG, Altenberger Straße 69, 4040 Linz, Austria; norbert.huber@recendt.at (N.H.); bernhard.reitinger@recendt.at (B.R.)

**Keywords:** Wire and Arc Additive Manufacturing (WAAM), fusion welding, NDT, laser-ultrasonic, laser-induced breakdown spectroscopy, laser opto-ultrasonic dual detection, thermography, acoustic emission, airborne optical microphone

## Abstract

In Wire and Arc Additive Manufacturing (WAAM) and fusion welding, various defects such as porosity, cracks, deformation and lack of fusion can occur during the fabrication process. These have a strong impact on the mechanical properties and can also lead to failure of the manufactured parts during service. These defects can be recognized using non-destructive testing (NDT) methods so that the examined workpiece is not harmed. This paper provides a comprehensive overview of various NDT techniques for WAAM and fusion welding, including laser-ultrasonic, acoustic emission with an airborne optical microphone, optical emission spectroscopy, laser-induced breakdown spectroscopy, laser opto-ultrasonic dual detection, thermography and also in-process defect detection via weld current monitoring with an oscilloscope. In addition, the novel research conducted, its operating principle and the equipment required to perform these techniques are presented. The minimum defect size that can be identified via NDT methods has been obtained from previous academic research or from tests carried out by companies. The use of these techniques in WAAM and fusion welding applications makes it possible to detect defects and to take a step towards the production of high-quality final components.

## 1. Introduction

In 1925, Backer introduced a novel technology named Wire and Arc Additive Manufacturing (WAAM) [1,2]. WAAM also known as Shape Metal Deposition (SMD), Shape Welding (SW) and Shape Melting (SM) belongs to Directed Energy Deposition (DED) based on ASTM F2792-12a [3]. It combines arc welding technologies, such as Gas Metal Arc Welding (GMAW), Gas Tungsten Arc Welding (GTAW) and Plasma Arc Welding (PAW), and wire materials to manufacture near net shape metallic components via a layer-by-layer deposition approach [4,5,6]. The difference between WAAM and welding is the geometry of the component and resultant effects on the temperature distribution. WAAM is assumed to be one of the most promising AM techniques in various industries, such as the aerospace, space and marine industries, owing to its capability of manufacturing complex and large components, high deposition rate and reduced wasted material and lead time resulting in cost reduction [2,7,8,9]. In general, the WAAM process is made up of the following steps: (i) creating a CAD model, (ii) slicing the 3D model into layers, (iii) generating an adequate deposition path, (iv) selecting proper welding parameters, such as, travel speed, current and voltage, (v) material deposition and (vi) post-processing [7]. Various materials, such as steel-, aluminium-, titanium- and Ni-based alloys are employed in WAAM [2]. In this technology, defects, such as porosity, cracking and oxidization may appear on the surface and subsurface of the final parts during the fabrication process as a results of higher heat accumulation in the part, in proper parameter configuration, contamination and inconstant weld pool dynamics [2,8]. These defects may cause the failure of the manufactured component; thus, it is extremely crucial to detect the defects in order to prevent failure during service [2]. The final welding properties are directly impacted by the welding parameters, such as voltage (polarity), current, travel speed, interpass temperature or preheat temperature. Control of the welding parameters makes it possible to enhance the properties of manufactured components [10]. Non-Destructive Testing (NDT) techniques have gained the interest of researchers due to their ability of detecting welding defects [11]. Despite traditional inspection techniques, the quality of additively manufactured or welded components can be evaluated cost efficiently and in real time using NDT methods so that the examined workpiece is not harmed [11,12,13,14]. Several NDT techniques are currently available, however, most of them are not applicable for real-time and automatic process monitoring for weld defect inspection in WAAM and fusion welding [9]. Kah et al. [11] investigated three different NDT techniques including eddy current, ultrasonic and real-time radiography and their types. It was reported that eddy current testing (ET) can detect small welding defects and discontinues in real time but only for conductive materials. Furthermore, deep welding imperfections cannot be inspected. In contrast, an Ultrasonic Testing (UT) technique covers these limitations and detects deeper welding defects in metals and plastics. Conventional UT transducers are not suitable for real-time and in-process monitoring of welded parts, as conventional UT transducers require contact with the specimen as well; they cannot withstand the very high temperatures of the WAAM and fusion welding processes. This limitation is overcome using laser ultrasonic [15]. Honarvar et al. [16] presented a review on distinct ultrasonic NDTs. The principle and their capability for in situ and offline inspection in additive manufacturing applications were discussed. Lopez et al. [9] reviewed various NDT techniques for WAAM applications in detail; however, the real-time acoustic emission inspection using an optical microphone as well as laser-induced breakdown spectroscopy were not discussed in their work. Table 1 provides additional information of reviewed NDT techniques by Lopez et al. [9] and Ricardo et al. [17]. The minimum detected defect size using each NDT method was extracted from the previous conducted academic research or from tests performed by companies. It must be pointed out that detecting the minimum size of the defects depends on various parameters, such as material (grain size, anisotropy, thermal conductivity), welding process, thickness of the sample and resolution of the equipment. These methods must be tested individually in order to acquire the best method for a specific application. 

The demand on reducing manufacturing lead times and increasing welding quality persuaded many researchers to develop available NDT techniques to inspect welding defects in real time and automatically [11]. Faramarzi et al. [18] merged image processing and radiographic non-destructive testing techniques to detect welding imperfections such as burn through, lack of fusion, lack of penetration and slag, automatically, as illustrated in Figure 1. Firstly, a program in MATLAB [19] was implemented in order to detect the burn through defects using data fusion and image processing. Then, the developed program was successfully applied for other above-mentioned welding defects. The developed image processing algorithm can be summarized in the following steps: (i) smoothing and thresholding, (ii) morphology operations, (iii) smoothing and (iv) boundary functions. Although radiography can be employed to automatically inspect all types of welding defects in complex welding geometries, very fine defects cannot be detected [11], and it also endangers human health [9]. Other image-based NDT, such as X-ray computed tomography and X-ray backscatter, are not suitable for real-time and on-line quality assessment as these require too much time to detect defects [9].

Seow et al. [20] employed dye penetrant testing, conventional ultrasonic testing and digital X-ray radiography to recognize crack-like imperfections in a WAAM Alloy 718 component. As depicted in Figure 2, these technologies are able to find crack-like defects in WAAM.

Wang et al. [21] employed 3D computed tomography technology and could find various defect types, including small spherical pores, inverted pear-shaped pores and cavities in a molybdenum WAAM component. Bento et al. [22] designed and examined an eddy current probe to detect flaws of an aluminium (AA 6082-T6) WAAM part during the manufacturing process. They revealed that intentionally made defects at a depth up to 5 mm and with a minimum thickness of 0.350 mm can be identified. In addition, the ability of defect detecting increases, as the number of coil turns rises. Lopez et al. [23] studied the ability of Phased Array Ultrasonic Testing (PAUT) to identify defects inside WAAM parts made of aluminium alloy (AA2319) with rough surfaces. They carried out some numerical simulations to determine the proper testing parameters and select an adequate transducer and then conducted some defect detection experiments. They reported that it was possible to detect defects sized between 2 and 5 mm. Chabot et al. [24] have demonstrated that PAUT is capable of detecting defects of 0.6 to 1 mm in aluminium WAAM components. Javadi et al. [12] applied PAUT and a total focusing method to detect artificially introduced tungsten carbide spheres with various diameters in a 20-layer wall made using WAAM. They were able to successfully detect most of the defects using a TFM technique. Later, Lukacs et al. [25] adopted Laser-Induced Phased Array (LIPA), full matrix capture data acquisition and a Total Focusing Method (TFM) to inspect a titanium alloy (Ti-6Al-4V) fabricated using plasma arc WAAM. As illustrated in Figure 3 and Figure 4, they clearly indicate that defects located at a depth of up to 10 mm inside the sample can be found offline by means of high-quality images.

In addition, it was found that the full matrix capture in LIPA is time consuming (nearly 14 min) due to the synthetic measurement approach including a large number of acquired A-scan signals, the physical scan of the laser and signal averaging. Therefore, it is currently not possible to implement this technique in real time and inline inspection. 

Each NDT technique has its own benefits and limitations and is able to detect specific defects and is used for specific materials. Thus, various NDTs must be combined to monitor the WAAM and fusion welding process. A lot of research has been conducted to investigate various NDT techniques in WAAM and fusion welding applications. Only little has been done on combining distinct contactless NDT techniques and sensor technologies in order to automatically detect welding defects in real time, monitor the weld pool and part characteristics, measure the temperature distribution and also process parameters. The main focus of this work is on WAAM. However, since most of the defects that can form during WAAM and fusion welding processing are similar (differences can result from different thermal fields and cooling conditions), non-destructive testing methods can also be applied to fusion welding. Therefore, in addition to WAAM, fusion welding is also covered in this paper. This review paper presents a concept of a combination of a set of contactless NDT and novel sensor technologies which possess the capability to be implemented and integrated in WAAM and fusion welding setups to automatically detect defects in real time and monitor the processes.

**Table 1 materials-15-03697-t001:** A review of NDT methods and the smallest detected defect for WAAM and fusion welding. (✓ represents suitable method and X stands for unsuitable technique).

NDT Method	Summary of the Operation Procedure	Suitable for Online/Offline Monitoring	The Smallest Detected Defect (µm)
Visual inspection [17]	An expert evaluates the workpiece with a naked eye or various simple equipment such as magnifiers or endoscopes [17].	X/✓	No information available
Liquid penetrant testing [9]	The fluorescent penetrant is applied on the surface of the material. It penetrates the defects, then the additional fluorescent is cleaned, and a developer used, which causes the defects to be identified [9].	X/✓	>750 [26]
Magnetic particle testing [9]	In the first step, component magnetization occurs. Imperfections cause a magnetic current to penetrate the material. After that, the particles are spread on the surface of the component, leading to particle accumulation in the penetration zone and, finally, welding defects detection [9].	X/✓ [17]	>1000 [27]
Eddy currents [9]	A magnetic field is created surrounding the examined workpiece by means of an emitted coil. The generated eddy currents inside the sample are alternated by the existing welding flaws. The welding defects can be detected via the variations in the impedance of the coil equivalent to the alternation of the eddy currents [9].	✓/✓ [17]	>350 [22]
Laser opto-ultrasonic dual detection [28,29]	It combines both laser ultrasonic and laser-induced breakdown spectroscopy technologies to detect defects and acquire elemental information of the tested material during the process [28,29].	✓/X	No information available
Conventional acoustic emission [9]	A piezoelectric transducer placed on the surface detects the generated acoustic waves during the manufacturing process [9].	✓/X [17]	No information available
Acoustic emission using optical microphone [30]	An airborne optical microphone with the ability to hear the frequencies up to 2 MHz is used to detect the soundwaves during the process [30].	✓/X [31]	No information available
Conventional ultrasonic testing [9]	Acoustic waves generated by a transducer, which has contact with the sample, are propagated into the specimen. These waves interact with the welding defects and then return to the surface of the specimen. These waves are detected and evaluated to recognize the defects [9].	X/✓ [9]	>500 [32]
Phased array ultrasonic testing [9]	A PC is employed to control each multi-element probe instead of single element probe in conventional ultrasonic testing to create a concentrated ultrasonic beam, and a software to direct it [9].	✓/✓ [17]	>600 [24]
Immersion ultrasonic testing [9]	In comparison to conventional ultrasonic, the examined component is plunged into the liquid (usually water). Using this technology eases the transmission of the waves into the sample [9].	X/✓ [17]	>500 [32]
Electro-magnetic acoustic transducer [9]	An electro-magnetic sensor is employed near to the surface of the sample to generate and capture the acoustic waves. This technology is contactless and does not require any couplant [9,33].	✓/✓ [17]	>500 [32]
Laser ultrasonic testing [9]	The excitation and reception of the soundwaves occurs by means of two different lasers [9].	✓/✓ [17]	>100 [34]
Radiographic inspection [9]	Although the sample uniformly receives the excited radiation energy, imperfections, density alternation and thickness areas captured the radiation energy ununiformly. Thereafter, film(s) or electronic devices are used to capture the absorption differences [9].	X/✓ [17]	>45 Digital Radiographic [35]
Real-time radiography (RTR) [36]	Compared to conventional radiography, digital data are generated during X-ray penetration in the sample [36].	✓/✓ [11]	>250 conventional RTR [11]
			>250 RTR with Image Processing [11]
			>50 Microfocus RTR with Image Processing [11]
X-ray backscatter [9]	One of the main comparison between the X-ray backscatter and conventional X-ray technique is that the returned X-ray energy from a single side of the tested sample is recorded in the X-ray backscatter technique [9,37].	✓/✓ [17]	>20 [9]
Computed tomography [9]	A number of 2D X-ray images are captured surrounding a rotation axis. These are collected and used to create a 3D model of the sample by applying algorithms [9,38].	X/✓ [17]	>600 [39]
			>10 for micro-CT [40]
Infrared Thermography [9]	During the monitoring, an IR camera is used to measure the temperature difference on the surface of the sample caused by the presence of the defects [9].	✓/✓ [17]	>400 [41]
Eddy current thermography [9]	The heat is generated in the examined material generated by eddy current method and recorded by an IR camera [9].	✓/✓ [9]	>400 [41]
Vibrothermography [9]	The produced soundwaves by an UT transducer inside the material collide with the defects and cause a heat release as a consequence of friction. Then, the released heat is captured via an IR camera [9].	X/✓ [9]	>400 [41]
Laser thermography [9]	The sample is heated up using a laser. The energy interacts with the defects. Assessing the heat distribution surrounding the laser spot on the surface of the material allows the defects to be identified [9].	✓/✓ [9]	>400 [41]
Voltage and current evaluation	During WAAM and fusion welding processes, voltage and current are captured in real time and/then analysed by means of statistical analysis tools or machine/deep learning techniques to detect defects [42].	✓/X	No information available
Optical emission spectroscopy [17]	The electronic temperature profile is determined during the process by means of assessing the generated light during welding process. This electronic temperature profile is then correlated with existing flaws in the component [17].	✓/X [17]	No information available

## 2. Laser-Ultrasonics Testing

The oscillation/space displacement of particles, their position restoring forces and their linkage to the surrounding particles which in turn leads to an energy transport is known as elastic or sound waves. The elastic waves typically used in non-destructive testing are so-called ultrasonic waves characterized by frequencies higher than the range audible to humans (>20 kHz) and therefore cannot be heard. However, they can be detected using different types of transducers. Transducers transform these elastic waves into electrical signals that can be monitored as visual signals on a monitor [43] and further analysed. The most commonly used ultrasound transducer technology is based on the piezo effect where stress leads to electric charge. A number of further technologies are available such as Electromagnetic Acoustic Transducers (EMAT), Capacitive Micromachined Ultrasonic Transducers (CMUT), magnetostrictive transducers and Laser Ultrasound (LU) to both generate and receive ultrasonic waves [16]. In this section we are going to focus in more detail on the contactless NDT technology LU. LU is a non-contact NDT technique that is typically composed of two systems, a pulsed laser (q-switched ns-laser) to excite elastic waves and a long pulsed or Continuous Wave (CW) laser in combination with an interferometer to detect the surface movement caused by these elastic waves [44,45]. It can be used in automation processes and harsh environments [46]. As shown in Figure 5, the LU test system consists of the mentioned units including electronic data acquisition and processing hardware [47].

A number of different propagation types (modes) of elastic waves can be used in UT inspection. Table 2 illustrates the most relevant propagation modes in solids. Generally, elastic waves can be categorized into bulk waves and guided waves. Bulk waves are divided into longitudinal (compression) or transverse (shear) waves. Guided waves own various modes, such as surface (Rayleigh) waves, plate (Lamb including Zero Group Velocity (ZGV) modes) waves [16,49,50,51]. Typically for LU, these waves can be excited and propagate simultaneously in the material and can further be distinguished via data processing and prior knowledge on the physical nature of the elastic wave. These waves can be generated either via a thermoelastic mechanism at lower laser energy [52] or ablation of the component via vaporization of a small amount of the upper layers at higher laser energy.

Figure 6 shows a symbolic sketch of two different ultrasound wave excitation phenomena: thermoelastic and ablative. Thermoelastic wave generation is a non-destructive method, in which a short-pulsed laser, with an energy density lower than the damage threshold of the material, excites a short pulse on the surface of the material. This causes heating, including thermal expansion of the surface of the material, in a very short time, which leads to particle motion and therefore elastic waves. Ablative generation is enabled by enhancing the laser energy density higher than the damage threshold of the material surface. A small amount of material is vaporized, which causes a backdraft to the surface, and this generates elastic waves revealing much higher amplitudes. The most prominent difference between the two ultrasonic excitation phenomena are different elastic wave amplitudes, different wave dispersion behaviours and being destructive and non-destructive [53,54]. In the next step, the excited elastic waves are detected by the second laser optical system [16,47,48,55,56], which in the majority of cases illuminates a certain spot on the material surface. The elastic waves generate small vibrations of the surface which lead to a phase modulation (Doppler shift) of the reflected laser light of the detection system. This is further demodulated to an amplitude modulation via the use of different interferometer systems (homodyne, heterodyne, self-interference, two beam, photorefractive, time delay).

LU enables competitive advantages compared to conventional ultrasonic NDT techniques, such as the ability to operate contactless, in real time, at various distances, even more than 1 m, and at any temperature, to measure the thickness, to inspect flaws and to characterize the material even on non-stationary components [45,47,58]. Furthermore, it requires no couplant between the transducer and the component, and it can operate at a very high bandwidth (from 1 MHz to 100 MHz or even more) [58]. Cracks, lack of fusion, porosity and residual stresses can be detected via UT in powder- and wire-based DED processes [16]. In addition, it has the capability to detect defects larger than 100 μm and up to 700 μm deep [9]. Many researchers focused on LU to implement it in AM applications because of its ability to contactlessly detect defects. Levesque et al. [59] investigated the use of laser ultrasound for offline defect analysis of INCONEL718 and Ti-6AL-4V coupons and compared the results with X-ray tomography. The coupons were processed using a laser powder, a laser wire and an electron beam wire deposition process. Laser ultrasound results were reconstructed using the Synthetic Aperture Focusing Technique (SAFT) method, improving resolution and the Signal-to-Noise Ratio (SNR). Defects such as porosity with typical sizes of about 0.4mm and lack of fusion in the laser wire deposition process could be clearly identified. It should be noted that the measurements were made from the bottom side of the coupon. Use on a rough surface for a later inline application still has to be tested. To overcome this, Levesque et al. [60] showed in a later publication the possibility of using LU for defect analysis of thick welded structures (butt welds). Since similar surface roughness and geometry problems are expected here as in the previous investigations, a comparison can be made to the applicability of LU to WAAM structures. LU was applied directly to the machined surface, and the reconstruction of the data using SAFT was corrected for the geometry information obtained via a profile camera. In this way, artificial defects (EDM slits in one) with a size of 2–3 mm could be resolved at a depth of 50 mm. In contrast to the works presented so far, Klein et al. [61] researched the possibility of using surface acoustic waves (Raleigh waves) to detect defects positioned near the surface. The use of SAW waves in contrast to the bulk wave approach should offer advantages in his explanations especially for near-surface defects. The analysis of the Surface Acoustic Wave (SAW) waves and their temporal displacement, induced by defects, was carried out in the work by means of wavelet analysis and numerical simulations. Artificial defects (flat bottom drilled holes, 1 mm diameter and 0.4 mm deep) were clearly detected on titanium and steel samples with machined surfaces. Following studies have to show the applicability on typical WAAM components with their relatively high surface roughness. A similar approach was taken by Dixon et al. [62], based on a pulsed laser to excite SAW waves and an EMAT (Electromagnetic Acoustic Transducer) detector. The combination of laser and EMAT allowed on the one hand a low-cost detector (in contrast to the typical laser-based LU detectors) and on the other hand the use of EMAT on materials with low electrical conductivity or magnetic properties. With this setup, samples with artificial defects and real components with porosities were investigated. The defects could be found, but in the conclusion, it is pointed out that a characterization of the defects in size and location is not possible at the time of publication, and therefore, this hybrid system could be used as a pre-screening technology. Zeng et al. [63] numerically and experimentally investigated three intentionally embedded defects in WAAM without any surface treatment: a crack with a width of 0.2 mm and a depth of 2 mm, a flat bottom hole with a diameter of 2 mm and a depth of 1 mm, and a through hole with a diameter of 2 mm and a depth of 2.5 mm. They successfully identified the abovementioned defects utilizing laser ultrasound technology. Fang et al. [64] determined that LU in transmission mode allows subsurface defects as small as 1 mm in diameter inside the additively manufactured A 316 L stainless steel part to be detected. The component was scanned simultaneously with a laser generator on one side and an ultrasonic detector on the other side of the probe. The time delay was measured using a cross-correlation algorithm. The position of the internal defects was evaluated based on the sample thickness and two maximum signal delay points at the left and right scanning. Guo et al. [65] merged a convolutional neural network (CNN) and wavelet transform techniques to evaluate the laser ultrasound signals and automatically detected the width of the subsurface defects artificially introduced into an aluminium alloy (AW 2024) plate. The laser ultrasound signals were transformed into images using the wavelet transform method. Afterwards, the transformed images were adopted as training data for the CNN approach. They proved that the applied technique is an effective methodology with a very high detection accuracy. Nomura et al. [15] assessed the feasibility of laser ultrasonic technology for real-time defect detection in GMAW of a mild steel. Two defects were investigated, including a lack of penetration and a solidification crack. For this purpose, a pulsed laser with a pulse width of 9 ns and a frequency of 100 Hz, placed at a distance of 50 mm behind the welding torch and 4.5 mm behind the melt pool, generated the sound waves in ablation mode, and a laser detector received the excited sound waves. The synthesis Aperture Focusing Technique (SAFT) [55,60,66,67] was adopted to acquire images by means of longitudinal sound wave velocities. Although the depth position of the defects was estimated inaccurately (around 5% deviation) due to the difference between the applied room temperature and the true value for the ultrasonic waves, it was successfully shown that LU is capable of detecting defects in real time as well as during the welding process. In addition, it was proposed that the heat diffusion caused by the welding process should be taken into account to overcome this deviation. Wei Zeng et al. [68] performed a Finite Element Method (FEM) simulation to study the interaction of generated soundwaves using laser ultrasonic with the subsurface defects in an aluminium sample. It was claimed that subsurface defects depth alternation impacts the maximum displacement of the echo and oscillating waves. Karabutov et al. [69] reported the detection of subsurface stress distribution for titanium and nickel alloys using LU.

## 3. Acoustic Emission

Acoustic Emission (AE) belongs to the NDT techniques which enable detection of welding imperfections and metallurgical transformations in a component under use or stress cost effectively. As these phenomena happen inside a material, a specific amount of energy discharges rapidly resulting in the generation of transient elastic waves. The elastic waves travel within the material to the surface and can be detected in real time using sensors, e.g., a piezoelectric transducer in conventional AE, mounted on the surface of the tested material or via vibrometers [42,70]. Transducers convert the generated mechanical motion on the surface into the electrical signal. Then, a Low Noise Preamplifier (LNA) is applied to increase signal amplitude and possibly reduce electrical noises (bandpass filtering). The signals are collected, post processed and further analysed. The AE frequency ranges typically from 150 up to 300 kHz [43,71,72,73,74,75]. AE detects the generated soundwaves during the process and therefore is a passive technology. In contrast, the ultrasonic testing method generates the soundwaves using an external soundwaves generator, such as a piezoelectric transducer (or a laser in LU) and consequently is an active technology. Conventional AE is not adequate for real-time inspection of WAAM and fusion welding processes [9]. Figure 7 indicates a graphical illustration of a conventional AE measurement system.

Besides piezoelectric transducers, wide-band transducers and optical microphones are employed in AE [74,77]. A microphone is a sensor that has been applied since the last century to detect pressure waves (sound waves) in gas or fluids. It is made up of a solid diaphragm and a displacement transducer and may be equipped with some extra components, e.g., mufflers and focusing reflector. It converts sound waves into electrical signal via its diaphragm vibrations caused by soundwaves collision. However, this feature constrains standard microphones to detect high frequency airborne sound waves intrinsically via the mechanical construction (leading to oscillations and high frequency damping) [78,79]. To overcome this limitation, commercial systems based on laser interferometry provide real-time, non-contact detection of sound waves in air with a variable frequency from 10 kHz up to 2 MHz using a laser interferometer replaced by a vibrating diaphragm in conventional microphones [30]. Figure 8 shows a sketch of the membrane-free optical microphone. The interaction of the ultrasonic waves and air results in a change of density and consequently the optical refractive index of the air. As a result, the wavelength of the laser beam, which is trapped inside the tiny laser interferometer, is affected. The intensity of the returned laser beam is obtained using a photodiode [77,80,81].

Various research has been conducted in order to gain deeper knowledge about the AE technology. Ramalho [82] et al. explored the impact of different impurities on the sound waves recorded by a microphone during the WAAM process by means of power spectral density and Short Time Fourier Transform (STFT) analysis techniques. They were able to successfully detect defects. Aboali et al. [83] employed energy, number of counts and amplitude of the AE signals to detect pre-introduced lack of fusion, porosity and slags in a carbon steel welded part. The results were compared with those of a defect-free part. It was demonstrated that the lowest values of energy and amplitude and the highest value of number of counts was registered for the defect-free part. The registered AE parameters were mostly impacted by slag defects followed by porosity and lack of fusion. The ability of AE to detect welding defects, including slag, porosity and crack was studied by Droubi et al. [84]. It was clearly evidenced that defect detection is much more straightforward using AE energy, Root Mean Square (RMS) and peak amplitude parameters. The recognition and detection of welding defects was precisely conducted via wavelet transform results. In addition, it was claimed that the distance between the sensor and the model clearly impacts the results. Luo et al. [85] applied the AE count statistic, Root Mean Square (RMS) waveform calculation and power spectrum distribution methods to analyse the AE signals during pulsed YAG laser welding. They claimed that plasma plume produces the recoil force and the thermal vibration. These are the source of the generated AE waves during pulsed YAG laser welding. Grad et al. [86] registered the produced AE waves using a microphone and a piezoelectric sensor during a GMAW process. It was stated that short circuiting and arc reignition are the main sources of generated acoustic waves during the GMAW process. Furthermore, the acoustic parameters are affected by the type of applied shielding gas and a wire extension length of greater than 12 mm. Zhang et al. [87] used acoustic emission and air coupled ultrasonic testing to study in real time the presence of burn through in GTAW. As shown in Figure 9, it was proved that when welding defects appear, a sudden surge in the AE absolute energy occurs. 

Lee et al. [88] investigated the generated plasma in CO_2_ laser lap welding using a photodiode and microphone. It was shown that as the pores and spatter formation amount increase, the signal intensity decreases.

The data analysis of the AE results, on which the subsequent interpretation is based, refers in most cases to the so-called detection of “signal events”. This means the characterization of acoustic signals, which exceed a certain amplitude limit, on number of counts (statistics), amplitude energy, spectrum distribution, etc. An exciting new area for the analysis and interpretation of AE signals is the use of machine (deep) learning methods. Here, some very interesting approaches similar to the process of LPBF (Laser Powder Bed Fusion) have already been made and could also be transferred/applied to WAAM. For further studies on this topic, we would like to refer the reader to the following non-exhaustive bibliography [89,90,91,92,93].

## 4. Optical Emission Spectroscopy

In Optical Emission Spectroscopy (OES) the electronic temperature profile is determined during the process by means of assessing the generated light during the welding process. This temperature profile is then correlated with existing flaws in the component [17,94]. Figure 10 illustrates a schematic of OES analysis.

OES has been discussed by a great number of authors in literature. Mills et al. [96] discussed the applicability of different emission spectroscopy techniques in GTAW. Mirapeix et al. [97] proposed a real-time technique according to the electronic temperature determination of the plasma to identify small defects in GTAW. Zhang et al. [98] applied optical emission spectroscopy for the first time to study the structural characteristics of WAAM of an Al alloy. It was proved that the spectral intensity, electron density and the width of the deposited layer are linearly related. The spectral intensity and the electron density rose with the increase in the number of deposited layers. These showed a steady state tendency, after which the number of deposited layers reached 5 to 7. In addition, it was evidenced that the porosity could be roughly identified via spectral analysis of the arc properties due to the relation between hydrogen content and porosity. Kisielewicz et al. [99] used inline optical spectroscopy to monitor laser blown powder directed energy deposition (LBP DED) of Alloy 718. It was found that spectroscopy can be a solution for inspecting the LBP-DED process for the deposition of Alloy 718, and it was also reported that laser power variables and continuum radiation intensity do clearly correlate. Nassar et al. [100] combined OES, data acquisition and a control system to recognize the predefined defects in real time during DED of a Ti-6Al-4V part. It was demonstrated that a correlation exists between atomic titanium (Ti I) and vanadium (V I) emissions and predefined defects in the part.

## 5. Laser-Induced Breakdown Spectroscopy

Laser induced breakdown spectroscopy (LIBS) is a real-time chemical analysis technology that can determine the qualitative and quantitative chemical composition of a material based on the wavelength and spectral intensity emitted by a laser-induced plasma. As shown in Figure 11, a pulsed laser, focusing optics, light collection optics, a spectrometer and a computer system are typically required to perform LIBS analysis. Usually, LIBS uses a Q-switched Nd:YAG laser that emits a short laser pulse (commonly with a pulse duration of 5–100 ns) which is focused onto the surface of the specimen. Owing to the high energy of the laser beam, a tiny amount of the sample is vapored and forms a plasma vapor cloud above the surface. The light emitted by the plasma is captured and transmitted into a spectrometer via an optical fibre, where the measurement of spectral intensities takes place. The LIBS process is controlled, and the collected data are analysed via a computer system [56,101].

LIBS is widely applied in various industrial applications, e.g., for the inspection of solar cells to detect impurities. It provides several advantages over other analytical techniques, such as Inductively Coupled Plasma Atomic Emission Spectroscopy (ICP-AES), Atomic Absorption Spectroscopy (AAS), X-ray Fluorescence (XRF) and Energy Dispersive X-ray (EDX) spectroscopy, e.g., it requires no sample pre-treatment and little or no sample preparation. Furthermore, it is performed in real time, in situ, in the field and contactlessly. It also possesses some limitations, for instance, lower accuracy, matrix effects, and lower spectral intensities, due to the emission absorption of adjacent atoms around the generated plasma [56,101]. Later, Double-Pulse LIBS (DP-LIBS), multi-pulse LIBS and Hand-Held LIBS (HH-LIBS) were developed. In comparison to conventional LIBS, DB-LIBS applies two laser pulses in sequence with different parameters, and multi-pulse LIBS performs the analysis using several laser pulses to enhance the acquired intensities [56].

A series of recent studies has indicated that LIBS is a powerful tool for the in-process analysis of the welded and additively manufactured parts. Lednev et al. [102] investigated the LIBS method to detect laser welding process failures in situ. They tested three methods, which included passive detection of the weld pool and weld plasma emission, online LIBS sampling of the solidified hot weld and in situ LIBS measurements of the weld pool surface. Among them, in situ LIBS measurements successfully identified the defective zone from the non-defective area. Taparli et al. [103] examined in situ the LIBS during the GTAW welding process. It was shown that the Ar shielding gas flow and the welding arc plasma considerably influence the emission lines of chromium, nickel and manganese. There is a lack of knowledge related to the implementation of LIBS in WAAM and fusion welding applications.

## 6. Laser Opto-Ultrasonic Dual Detection

The Laser Opto-Ultrasonic Dual (LOUD) detection approach is based on the Laser-Induced Breakdown Spectroscopy (LIBS) and Laser Ultrasonic (LU) technologies. Figure 12 displays the LOUD technology. The process can be briefly explained as follows. A laser generates acoustic waves in ablation mode, the produced plasma spectra are collected via an optical collector and transmitted through a fibre into a spectrometer. The excited acoustic waves are received by an ultrasound detector and transmitted into a data acquisition card. A Digital Delay Generator (DDG) is provided to activate the laser, Data Acquisition (DAQ) card and Charge-Coupled Device (CCD) detector. Plasma spectra is used to detect elemental information, and the excited acoustic waves are recorded to investigate defects or residual stresses [28,29].

Figure 13 illustrates a possible concept of an online LOUD monitoring system proposed by Ma et al. [28], in which the ultrasonic detector and optical collector are attached to the robot arm. 

Ma et al. [28] successfully utilized the LOUD approach for the first time to determine residual stresses and elemental information, as well as defects of an aluminium alloy (A6061) produced with WAAM in-process and simultaneously. The use of LOUD as an NDT online monitoring tool in WAAM was recommended due to its time and cost efficiency. In another work by Ma et al. [29], they investigated at the same time the grain size and elemental distribution in an aluminium alloy produced in WAAM using LOUD. The results were in accordance with those obtained using Electron Backscatter Diffraction (EBSD). It was also reported that LOUD is an excellent tool for evaluating the mechanical and chemical characteristics of AM parts. There is lack of information regarding applying LOUD for other metals, such as titanium alloy and other aluminium alloy-based parts.

## 7. Thermography

Thermography is commonly employed in many fields as it is able to detect subsurface faults, determine thermophysical properties and measure the coating thickness of the components [104]. There are various criteria for the classification of thermography, e.g., testing approach and applied stimulation source. As shown in Figure 14, thermography is divided into two main techniques according to the testing approach: (1) active thermography and (2) passive thermography. Based on the applied stimulation source, active thermography is subdivided into five categories, including, Lock-In Thermography (LIT), Pulsed Thermography (PT), Vibro-Thermography (VT), Step Heating Thermography (SHT), and eddy current thermography [105,106,107].

Figure 15 indicates an illustration of defect detection by means of the thermography method. Figure 16 shows a schematic of the active thermography technology. In active thermography, the under-inspection component is exposed to thermal stimulation (cold or warm) from an external source, such as halogen lamps, pulsed lamps and laser. However, in passive thermography, the heat distribution inside the material is inherent, e.g., as a result of the manufacturing process. The homogenous heat distribution inside the material interacts with the imperfections within the material. This inhomogeneous heat diffusion on the surface of the part caused by existing defects inside the material is recorded via an infrared (IR) camera. The IR camera receives the emitted infrared waves from the surface of the material and converts them into electrical signals and subsequently into IR images. Then, the defects can be visualized by analysing the images [105,106,107].

Lock-in thermography: It belongs to NDT techniques according to photothermal radiometry, in which one or some heat sources are used to continuously heat the surface of the part under inspection. The heat waves are transmitted to the components though radiation. They interact with the imperfections within the part and return. An infrared camera is used to capture the returned heat waves and a lock-in amplifier measures the amplitude and phase of the modulation. The phase and amplitude of the returned thermal waves change owing to the difference of the thermal properties of the defected and non-defected regions inside the component [106,109]. Data acquisition takes more time in lock-in thermography [110].

Pulsed Thermography (PT): It has been employed in aerospace industry for various applications, such as for the evaluation of airplane components [104]. This technology is able to detect online and very quickly various defects such as cracks, fatigue damage, rust and delamination in different materials [111]. Infrared lamps, halogen lamps and hot air guns are used to generate one or more thermal pulse(s) on the surface of the components within 2–10 ms. Heat transfers inside the material and interacts with the existing defects. The inhomogeneous heat distribution on the surface of the part caused by the internal defects is measured using an IR camera. Figure 17 illustrates a common PT setup. An external heat generator (infrared lamps, halogen lamps, hot air guns), an infrared camera, a control unit and a computer with data processing software are usually the required equipment to perform PT [111,112].

Step pulsed thermography: Compared to pulsed thermography, step pulsed thermography applies longer, homogenous and uninterrupted pulses with a low energy density to detect defects located at a deeper distance from the surface of the material. Defects have a different temperature and heat transfer rate in comparison to non-damaged parts of the under-inspection component. An IR camera is used to record the whole heating and cooling process in step pulsed thermography. The surface temperature alternations are captured and analysed to detect defects [113,114].

Vibrothermography: It combines an ultrasonic testing method and thermography to detect subsurface and near-surface defects, such as cracks, disbands and delamination quickly and precisely. As depicted in Figure 18, the soundwaves created by a vibration source, e.g., a piezoelectric transducer, travel through the component. As they meet the defects, the vibration energy is converted into heat due to friction. The generated heat is transferred to the surface and captured via an IR camera. A common setup for vibrothermography is composed of an ultrasonic vibration source, an infrared camera, a control unit and a PC with a data processing software [112,115].

Eddy current thermography: This technology is categorized under non-optical thermography methods [106] and takes advantage of eddy current and thermography NDT approaches. Figure 19 displays an illustration of eddy current thermography. Eddy current flow is created inductively inside the electroconductive materials. This results in an increasing temperature of the component. When defects exist inside the material being tested, the current flow and subsequently the heat distribution is disturbed. The non-uniform heat distribution is captured through an IR camera [117]. Eddy current thermography makes it possible to quickly and contactlessly inspect the components with a high resolution [110].

Thermography has been widely researched to detect various welding defects in-process in real time. Bacelar et al. [17] utilised passive thermography to detect four intentionally introduced holes during the WAAM process. It was found that the defective area has a higher temperature compared to the flawless region as it dissipated less heat to the surroundings. Yang et al. [119] monitored the surface temperature of the deposited layers in the WAAM process by means of passive thermography and reported that thermography is able to precisely measure the surface temperature during the process. Mireles et al. [39] explored the ability of the IR thermography to monitor in situ the parts produced by powder bed fusion technology. They purposely placed different defects with diverse sizes and shapes into an assembly part. An IR camera with a resolution of 640 × 480 and a pixel length of 260 µm was adopted to capture the component under inspection. It was reported that defects smaller than 600 µm were not detected with the setup used. Runnemalm et al. [120] evaluated three types of excitation sources, including a flash lamp, eddy current induction and a continuous laser for thermography to study surface defects such as cracks, pores and lack of penetration. A combination of a flash lamp positioned at a distance of 120 mm with 6 kJ energy within 0.05 s, a FLIR SC5650 IR camera with a spectral range of 2.5–5.1 µm and an optical lens of 27 mm allowed a notch to be detected with a length of 760 µm and a width of 400 µm. Five holes with a diameter of 1.0 to 2.5 mm were created on the workpiece, which were examined via eddy current thermography. An induction coil located at a distance of greater than 10 mm from the component heated up the component. All holes were recognized via eddy current thermography. It was possible to detect all eight artificially produced defects on the component under evaluation by means of the laser. Roemer et al. [112] drew a comparison between laser thermography and vibrothermography for fatigue crack detection in an aluminium bar. They coated the sample surface with black paint due to the fact that aluminium has a lower heat emissivity. They applied a laser power of 100 W with a pulse length of 100 ms to generate a 5 K temperature increase and to perform the laser thermography experiment. The heat diffusion was monitored with an IR camera with a resolution of 256 × 320 and a frequency of 60 Hz. In the second experiment, an ultrasonic transducer with a frequency of 35 kHz and a power of 500 W generated sound waves with a pulse duration of 500 ms. The process was recorded with an IR camera with the same resolution as before and a frequency of 150 Hz. In both experiments, fatigue cracks were identified easily. They also claimed that an IR camera equipped with a zoom and focus lens enables the detection of micro size defects in laser thermography. Sreedhar et al. [121] utilized the passive thermography method for real-time inspection of a TIG-welded aluminium alloy-based tank (aluminium alloy 2219). An IR camera was attached on the welding arm at an angle of 60° to the welding plate and 150 mm behind the welding arc to capture a newly welded area of 100 mm. They could detect a cluster pore of size 0.6 mm × 0.4 mm by means of passive thermography. Elkihel et al. [122] studied the thermal propagation of a weld joint using an active thermography method. They heated the weld joint inductively up to 80 °C and captured the heat propagation on the weld joint using an FLIR T440 infrared camera with a resolution of 320 × 240 and a bandwidth of 7.5 to 13 µm pixels. It was claimed that the heat loss on the weld zone is much more significant than that of the defective region. Some researchers combined image processing and thermography techniques to ensure the quality of the weld seam. Massaro et al. [123] proposed a novel technique for identifying weld defects on a welded steel tank (AISI 304/316) using infrared thermography and image processing. They cut out a specimen and excited it with a heat gun. The heat distribution on the surface was registered using a FLIR T 1020 with a resolution of 1024 × 768 pixels. It was reported that a combination of IR thermography and various image processing techniques, including the line calculus method, 2D K-Means algorithm, 2D morphology functions and a Long Short Term Memory (LSTM) artificial neural network can be a powerful tool for real-time identification and classification of weld defects. Ziegler et al. [124] evaluated the use of high-power laser excitation sources in the lock-in thermography technique. They claimed that the use of high-power lasers instead of LEDs and halogen lamps has practically no influence on the thermal emission generated by the excited sample, as their emission is based on electroluminescence. Thus, it can be performed in single sided transient thermography. Furthermore, it is possible to use high-power laser arrays for the analysis of highly reflective materials, such as aluminium, if additional power scaling or focusing is provided. Cerniglia et al. [125] evaluated two additively manufactured Inconel 600 specimens with various intentionally inserted micro-sized defects located at different depths to establish an analogy between laser ultrasonic and laser thermography techniques. Both methods demonstrated their ability to inspect the samples in-line. They emphasized that for the deployment of laser thermography for automatic inline inspection, liquid-cooled IR cameras should be replaced by microbolometric IR cameras since they are more expensive and larger. In addition, using laser thermography for component surfaces with low emissivity values requires a high-power laser.

## 8. Defect Detection by Monitoring WAAM and Fusion Welding Process Parameters

Various physical phenomena, such as metal transfer, short circuiting, spatter, ionization, gas-metal reactions occur very fast during the welding process, which causes variations in current and voltage. Distinct welding parameters can be assessed, if the current and voltage signals are recorded simultaneously as these phenomena take place [126]. High speed data acquisition systems, such as digital storage oscilloscopes (DSOs) are able to register these signals [127]. 

DSOs have been widely used in the academic area to monitor welding parameters. Mičian et al. [128] utilised a digital oscilloscope in combination with other equipment such as a galvanic separator and a notebook with a software to capture the instantaneous current and voltage of the CMT process during MIG brazing of automotive components. Kumar et al. [129] employed a DSO with a sample rate of 40 KHz to collect the welding parameters for two inverter and two generator power sources during the welding using two different electrodes. It was claimed that a commercial DSO is able to obtain the welding data and can be further used to assess the welding process in situ and online and can be a powerful competitor to other data acquisition systems designed for the same purpose due to the comparable data acquisition speed, volume and accuracy. Savyasachi et al. [130] took advantage of a DSO and a high-speed camera simultaneously to monitor the Shielded Metal Arc Welding (SMAW). They proved that combining both technologies allows the investigation of different physical phenomena during the process at once. In addition, the collected data from a DSO require proper filtering to analyse the data. In another study conducted by Kumar et al. [126], it was found that Fast Fourier Transform Low Pass Filter (FFT LPF) is a suitable method to filter the acquired signals during SMAW due to its higher signal-to-noise ratio value compared to other techniques. In addition, a 100 KHz sample rate is a sufficient value for data acquisition using a DSO during SMAW. Mazlan et al. [131] successfully combined a DSO and a Short Time Energy (STE) method to detect welding defects in GMAW. The acquired welding current using a DSO was applied as input data for the STE analysis method conducted in MATLAB/Simulink. Using this method enabled the smooth current and disturbance current to be distinguished. This results in differentiating the defective weld from the non-defective weld. Šoštarić et al. [10] introduced an online monitoring system to acquire the main welding parameters and to process the data in resistance welding and arc welding. They made an analogy between the acquired results using a self-developed online monitoring system and those obtained using an oscilloscope. The results demonstrated that the self-developed online monitoring has the ability to be implemented in practical applications.

## 9. Summary and Conclusions

This paper provides a comprehensive review of various NDT techniques, including laser-ultrasonic, acoustic emission with an airborne optical microphone, optical emission spectroscopy, laser-induced breakdown spectroscopy, laser opto-ultrasonic dual detection, thermography and also in-process defect detection via monitoring the process parameters in WAAM and fusion welding. In addition, novel research results, operating principles and the equipment required to perform these techniques have been presented. 

The minimum detectable welding defect size of most current NDT techniques was collected via previous research or experience of companies. According to this review paper, the following conclusions can be drawn:LU is a fast technique and capable of detecting internal defects as small as 100 µm in WAAM and fusion welding. It requires no contact with the sample and can be implemented in harsh environments and also for the automation processes. However, LU vaporizes a small amount of the component under inspection.Acoustic emission is able to measure the soundwaves ranges from 150 up to 300 kHz during the manufacturing process. It is cost-effective and can be used for defect detection during WAAM and fusion welding. Since it is a passive technology, it cannot be used for offline monitoring. In addition, there is lack of knowledge on applying a membrane-free optical microphone for defect detection in WAAM and fusion welding.Laser induced breakdown spectroscopy is capable of acquiring elemental information of the sample and detecting defects such as porosity by means of assessing the chemical elements of the sample. Detecting other types of the defects in WAAM and fusion welding has not been investigated yet.Laser opto-ultrasonic dual detection can rapidly detect defects and elemental information at the same time during the manufacturing process without having contact with the sample.OES is a contactless technique and able to detect defects in situ. However, it is not appropriate for offline monitoring.Thermography is able to detect surface and subsurface defects and recognize flaws as small as 600 µm in WAAM and fusion welding. It requires no contact with the sample and can measure the thermophysical properties of the part online. It can be used as a signal for real-time closed-loop control systems, however, a heated sample and a proper machine learning algorithm for evaluation are required.Monitoring process parameters, such as voltage and current using DSOs or data acquisition systems enables real-time defect detection. These signals are fast and sensitive and suitable for real-time closed-loop control systems.

As each method possesses its own merits and demerits, a combination of different NDT methods is required to monitor the WAAM and fusion welding processes in real time and to ensure production of high quality and defect-free final parts. Future work should focus on the combination of the most suited NDT techniques at the laboratory scale to detect defects in real time during WAAM of light metal alloys.

## Figures and Tables

**Figure 1 materials-15-03697-f001:**
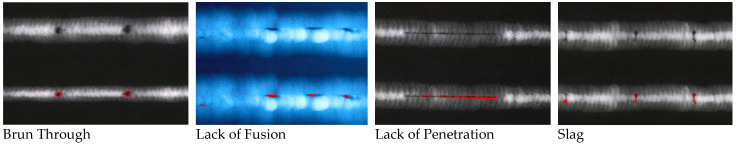
A combination of radiographic and image processing techniques for welding defect detection conducted by Faramarzi et al. [18].

**Figure 2 materials-15-03697-f002:**
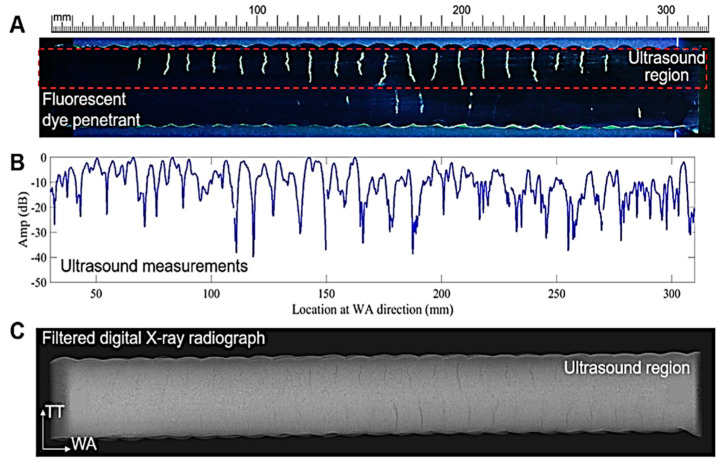
A comparison between dye penetrant testing (the dashed section in (**A**) depicts the area examined with ultrasound techniques shown in (**B**)) (**A**), conventional ultrasonic (**B**) and X-ray radiographic technologies (**C**) carried out by Seow et al. [20].

**Figure 3 materials-15-03697-f003:**
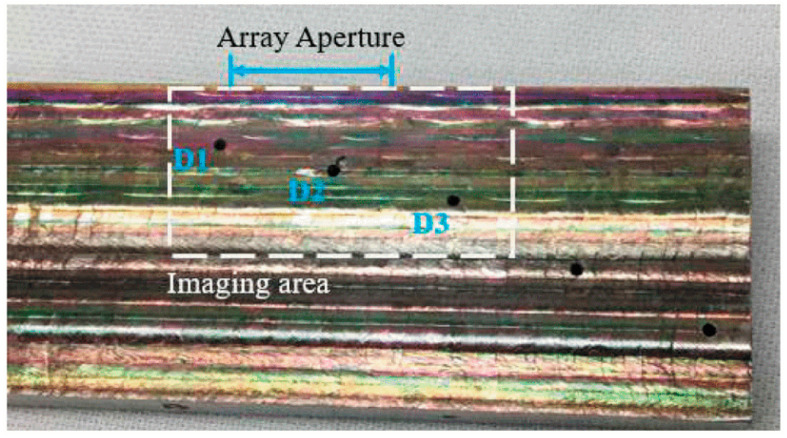
Inspected WAAM component. D1, D2 and D3 represent the welding defects [25].

**Figure 4 materials-15-03697-f004:**
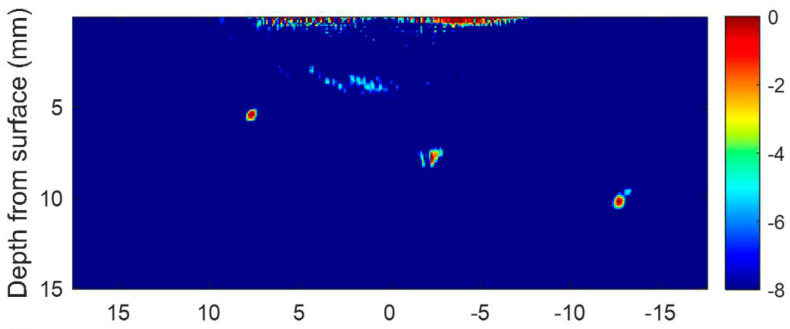
Intentionally introduced defects (D1, D2 and D3 in Figure 3) are detected by means of TFM image of the component using ultrasonic longitudinal waves [25].

**Figure 5 materials-15-03697-f005:**
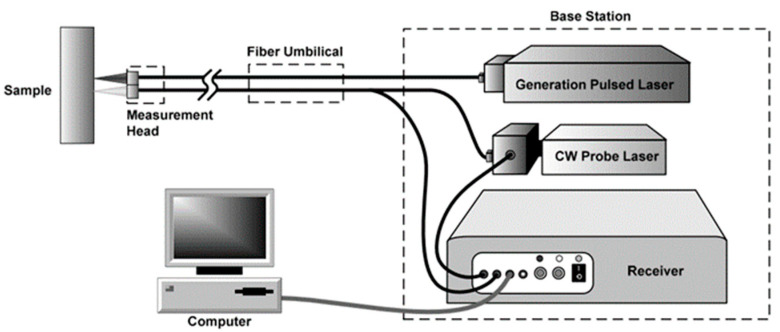
Laser ultrasonic inspection system [48].

**Figure 6 materials-15-03697-f006:**
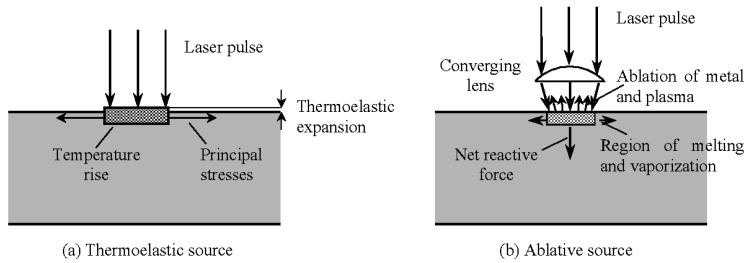
Thermoelastic (**a**) and ablative (**b**) phenomena in UL [57].

**Figure 7 materials-15-03697-f007:**
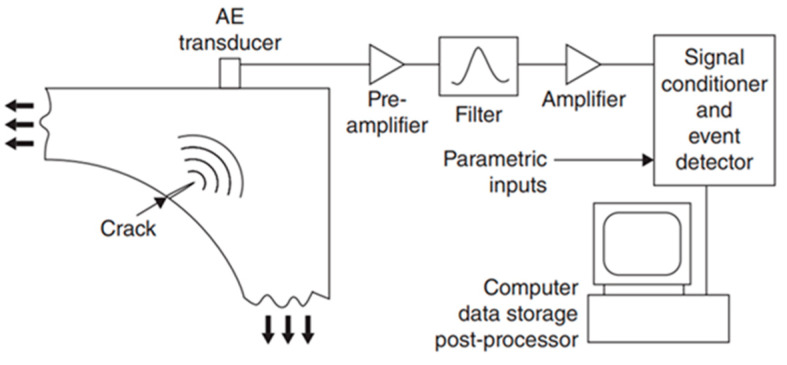
Graphical illustration of a conventional AE measurement system [76].

**Figure 8 materials-15-03697-f008:**
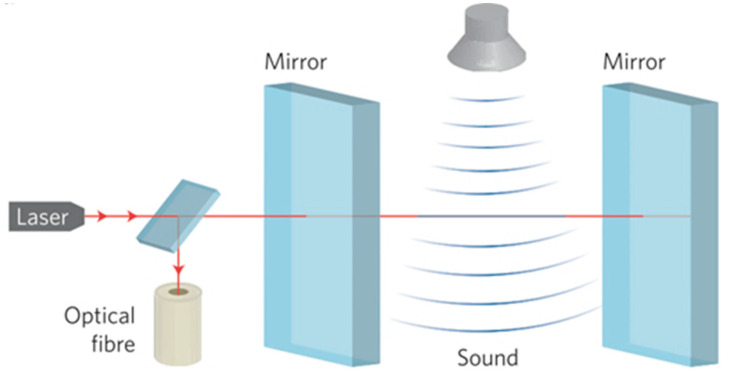
Schematic of the membrane-free optical microphone by Xarion [77].

**Figure 9 materials-15-03697-f009:**
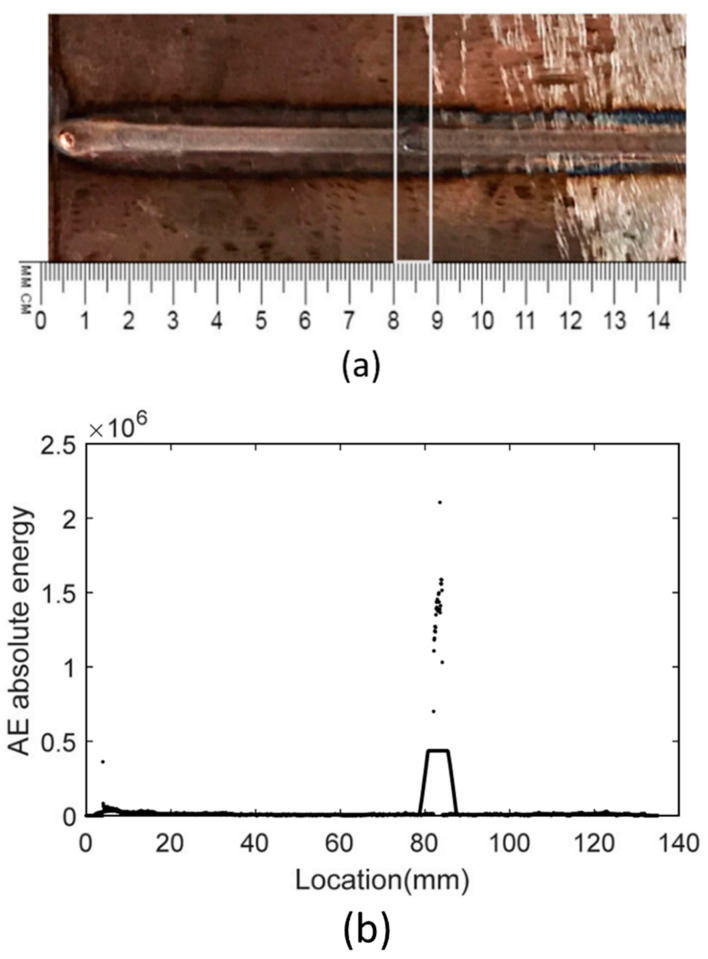
A correlation between acoustic emission absolute energy and welding defect [87]. (**a**) depicts the inspected sample. (**b**) shows the acquired AE absolute energy of the inspected weld seam.

**Figure 10 materials-15-03697-f010:**
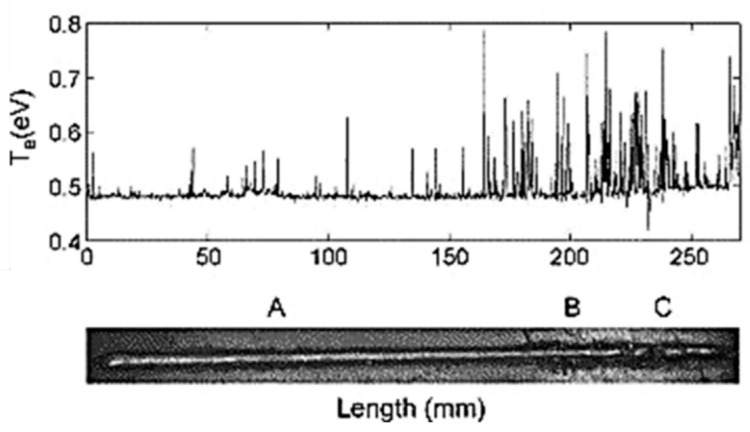
A schematic of the OES technique [95]. In this illustration, the recorded electronic temperature is correlated with the existing welding defects (A, B and C).

**Figure 11 materials-15-03697-f011:**
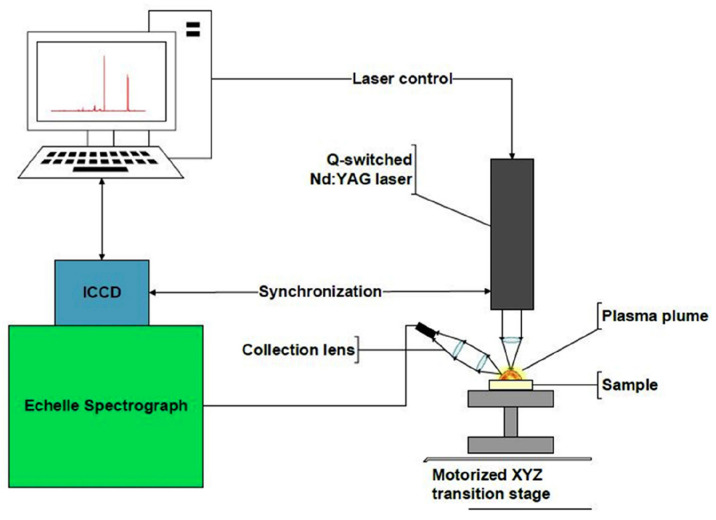
Picture of the LIBS setup [56].

**Figure 12 materials-15-03697-f012:**
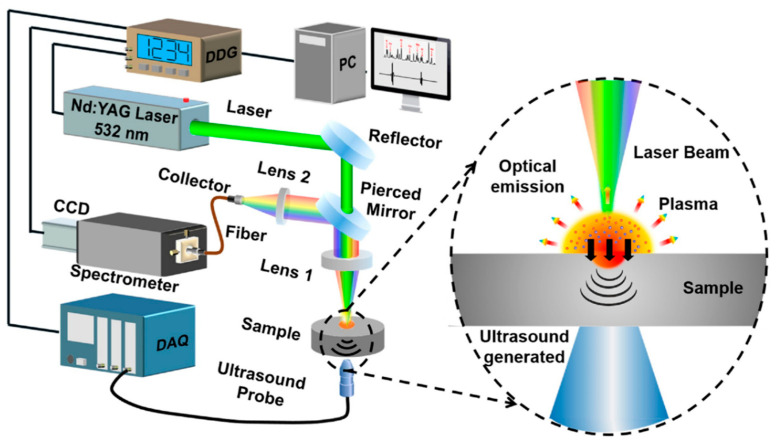
A picture of the LOUD process [28].

**Figure 13 materials-15-03697-f013:**
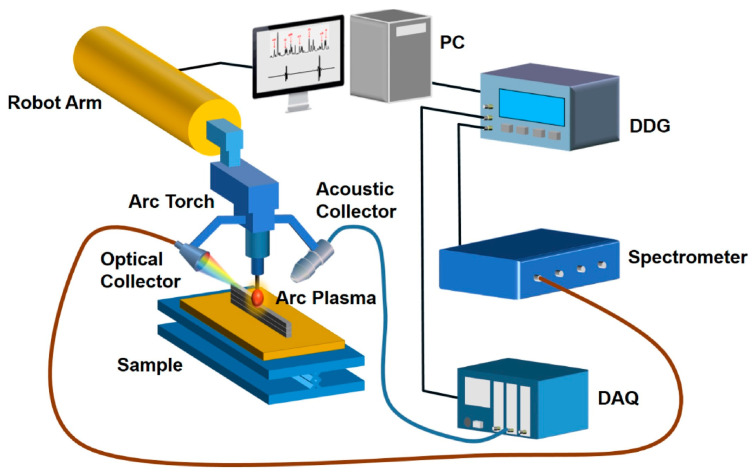
A schematic of a proposed laser opto-ultrasonic setup for online monitoring by Ma et al. [28].

**Figure 14 materials-15-03697-f014:**
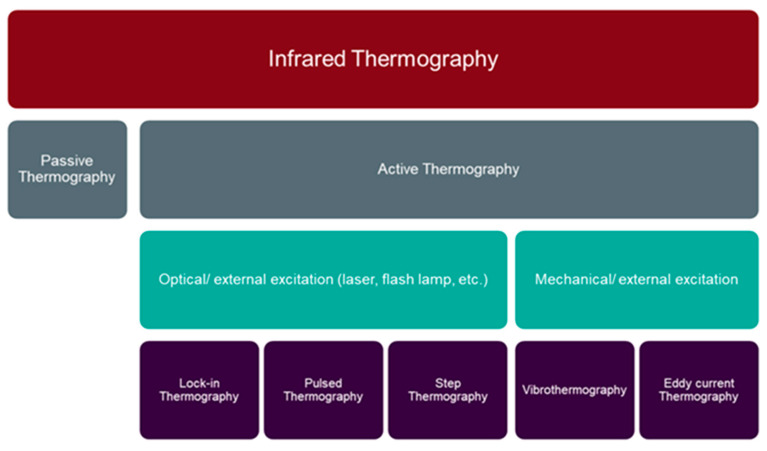
Thermography NDT techniques [105,106].

**Figure 15 materials-15-03697-f015:**
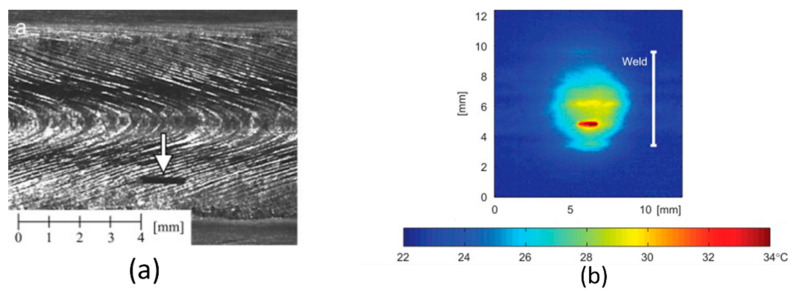
Defect detection by means of the thermography technology performed by Broberg [108]. (**a**) displays the defect (the arrow shows its position). (**b**) shows the captured thermal image, in which the defect can be recognised.

**Figure 16 materials-15-03697-f016:**
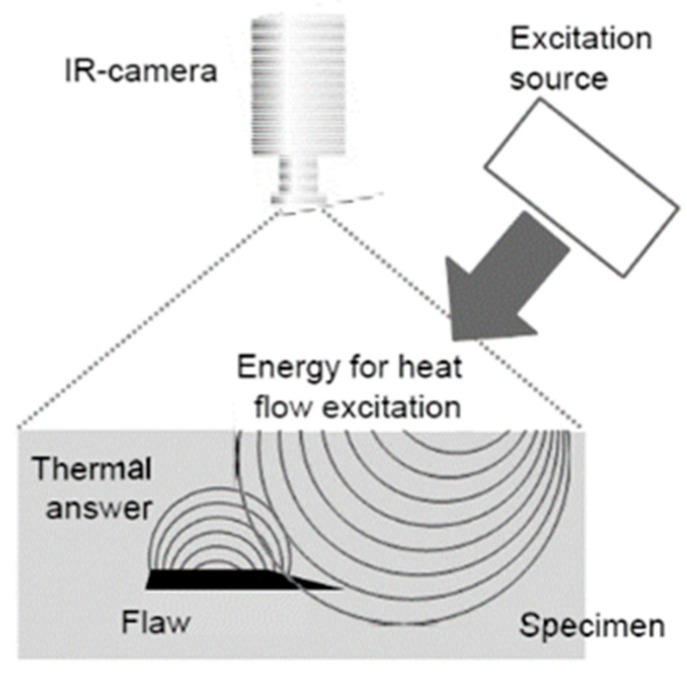
A schematic of active thermography [106].

**Figure 17 materials-15-03697-f017:**
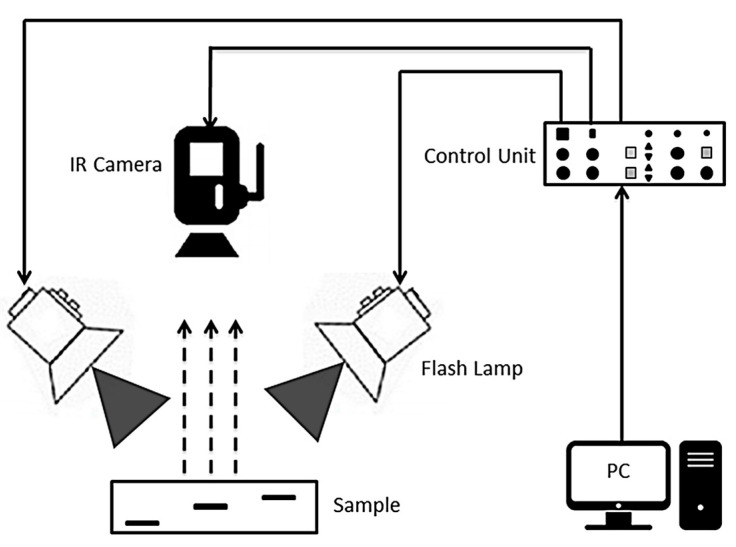
Pulse thermography setup [105].

**Figure 18 materials-15-03697-f018:**
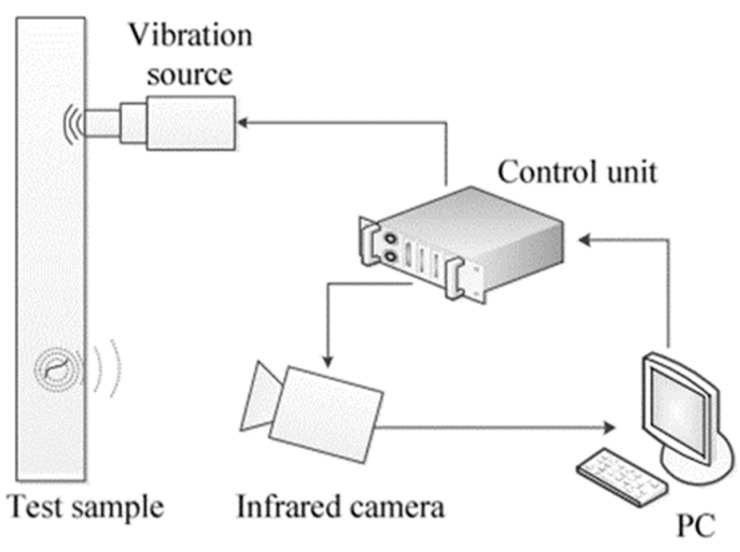
A picture of a vibrothermography setup [116].

**Figure 19 materials-15-03697-f019:**
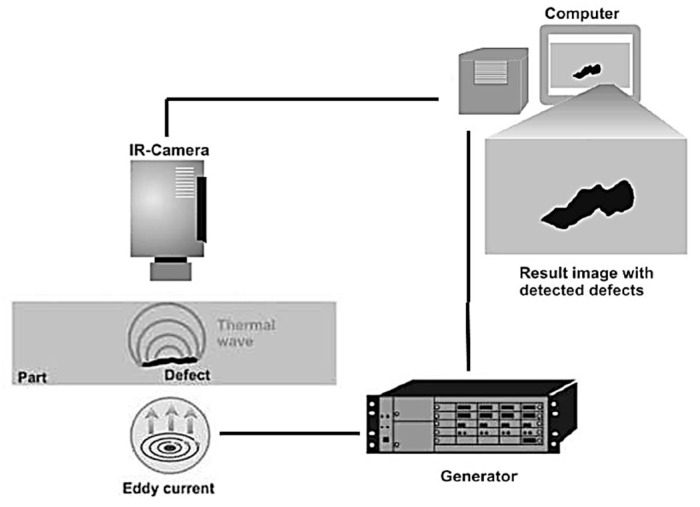
An illustration of eddy current thermography [118].

**Table 2 materials-15-03697-t002:** The most relevant propagation modes of ultrasonic waves in solids [49,50,51].

Propagation Mode	Description
Longitudinal (compression)	The particle motion is parallel to the wave travel direction.
Transverse (shear)	The particle vibration is perpendicular to the wave travel direction.
Surface (Rayleigh)	The wave is generated at the surface of thick solids caused by an elliptical motion of particles.
Plate (Lamb including ZGV modes)	A complex particle motion happens throughout the thickness and parallel to the surface of the material.

## Data Availability

The data can be found in the original works cited throughout this article.

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
