# Peer review of "A Review of Non-Destructive Testing (NDT) Techniques for Defect Detection: Application to Fusion Welding and Future Wire Arc Additive Manufacturing Processes"

_materials, 2022, doi:10.3390/ma15103697_

Round 1
Reviewer 1 Report
The authors review various NDT technologies applied in WAAM, including laser-ultrasonic, acoustic emission with an airborne optical microphone, optical emission spectroscopy, laser-induced breakdown spectroscopy, laser-opto-ultrasonic dual detection, thermography and also in-process defect detection by monitoring the process parameters. The WAAM community can understand the novel research results, operating principles, equipment, merits and demerits of these NDT technologies through this review, and may achieve the preparation of high-quality WAAM components by using these NDT technologies. Thus, I recommend publication of this review.
Author Response
Response to reviewers:
We would like to thank the reviewers for their excellent and fair reports and recommendations for improvement. All changes made to the manuscript are highlighted in yellow in the revised version of the manuscript and the comments are addressed each below:
Reviewer #1:
The authors review various NDT technologies applied in WAAM, including laser-ultrasonic, acoustic emission with an airborne optical microphone, optical emission spectroscopy, laser-induced breakdown spectroscopy, laser-opto-ultrasonic dual detection, thermography and also in-process defect detection by monitoring the process parameters. The WAAM community can understand the novel research results, operating principles, equipment, merits and demerits of these NDT technologies through this review and may achieve the preparation of high-quality WAAM components by using these NDT technologies. Thus, I recommend publication of this review.
The authors thank the reviewer for the careful consideration of the manuscript, and the recommendation for publishing the paper.
Masoud Shaloo
Junior Research Engineer | Center for Low-Emission Transport
LKR Leichtmetallkompetenzzentrum Ranshofen GmbH
Reviewer 2 Report
The article reviews contactless non-destructive testing for defect detection during the WAAM. The manuscript does not justify the title. The article fails to serve the purpose of a review article on several counts:
- The article does not review the NDT of WAAM. In the introduction, a few examples of NDT applicaion in WAAM are given then the rest of the article is related to the description of NDT techniques.
- In the name of WAAM, the cases of welding are referred. The actual number of articles related to NDT of WAAM and are referred to in this review are very few. Most of the articles are related to general NDT.
- Table 2, titled “A review of NDT methods and the smallest detected defect for WAAM”, is a tabular form of information picked from another review article. To make it look like a review, a reference to the broucher of NDT instruments picked from the internet is added. Ther are a general description of instruments and are not related to the WAAM
- What is the objective, the data collection policy adopted to review, and how information is classified and presented….nothing is there in this review?
- Section 2 to 5, which makes up 80% of the article, is a simple description of NDT techniques. These sections hardly relate to WAAM. They are a bunch of text parked.
- There are three sections there which are number 5. Due diligence is missing.
- Conclusions section does not make sense.
Author Response
Response to reviewers:
We would like to thank the reviewers for their excellent and fair reports and recommendations for improvement. All changes made to the manuscript are highlighted in yellow in the revised version of the manuscript and the comments are addressed each below:
Reviewer #2:
The article reviews contactless non-destructive testing for defect detection during the WAAM. The manuscript does not justify the title. The article fails to serve the purpose of a review article on several counts:
The authors would like to thank the reviewer for the detailed review of the manuscript and will definitely adjust the title according to the recommendation.
In order to meet the reviewer’s concerns regarding on justifying title of the paper, the article’s title was revised.
1). The article does not review the NDT of WAAM. In the introduction, a few examples of NDT application in WAAM are given then the rest of the article is related to the description of NDT techniques.
WAAM is a very young manufacturing technology. Therefore, several NDT techniques available have not been tested so far. However, the process is very similar to fusion welding, and we have therefore reviewed their application and suggested transfer to WAAM. We agree that in this regard an adaption of the title is required, as has been done in the revised article.
2). In the name of WAAM, the cases of welding are referred. The actual number of articles related to NDT of WAAM and are referred to in this review are very few. Most of the articles are related to general NDT.
The difference between WAAM and welding is the geometry of the component. However, defects that can be formed during processing are similar (Differences may arise from varying thermal fields and cooling conditions). Therefore, it is suggested that knowledge of the NDT techniques is transferable.
3). Table 2, titled “A review of NDT methods and the smallest detected defect for WAAM”, is a tabular form of information picked from another review article. To make it look like a review, a reference to the broucher of NDT instruments picked from the internet is added. There are a general description of instruments and are not related to the WAAM.
The authors are aware that this table has been based on the works of Lopez et al. and Ricardo, but it has been extended using information from various references as it can be seen from the reference section of the paper. We kindly disagree with the reviewer’s opinion regarding inclusion of information from the internet and brochures.
4). What is the objective, the data collection policy adopted to review, and how information is classified and presented….nothing is there in this review?
The objective is to provide a sound bases for future works on NDT applications for additive manufacturing. The objective has been outlined in the last paragraph of the introduction.
The data has been carefully reviewed prior to inclusion into this article by experts from WAAM technology and NDT technologies as it can be seen on the affiliation of the coauthors.
To the authors it remains a little bit unclear what the reviewer means by this comment. We would happily incorporate further suggestions after clarification.
5). Section 2 to 5, which makes up 80% of the article, is a simple description of NDT techniques. These sections hardly relate to WAAM. They are a bunch of text parked.
To the author’s opinion review articles address not only experts, but also unseasoned readers, therefore, additional information on each NDT technique has been provided.
6). There are three sections there which are number 5. Due diligence is missing.
We apologize for this error which has been caused by copying into the template.
7). Conclusions section does not make sense.
To meet the reviewer’s concerns, we have renamed this section ”summary“ as this title more accurately reflects the content of this section.
Masoud Shaloo
Junior Research Engineer | Center for Low-Emission Transport
LKR Leichtmetallkompetenzzentrum Ranshofen GmbH

Reviewer 3 Report
In paper Wire and Arc Additive Manufacturing (WAAM) is analise, various defects such as porosity, cracks, deformation and lack of fusion can occur during the fabrication process. These have a strong impact on the mechanical properties and can also lead to failure of the manufactured parts during service. These defects can be recognized using non-destructive testing (NDT) methods so that the examined workpiece is not harmed. This paper provides a comprehensive overview of various NDT techniques for WAAM, including laser-ultrasonic, acoustic emission with an airborne optical micro- phone, optical emission spectroscopy, laser-induced breakdown spectroscopy, laser-opto-ultrasonic dual detection, thermography and also in-process defect detection by weld current monitoring with an oscilloscope. In addition, the novel research conducted, its operating principle and the equipment required to perform these techniques are presented. The minimum defect size that can be identified by NDT methods has been obtained from previous academic research or from tests carried out by companies. The use of these techniques in WAAM applications makes it possible to detect defects and to take a step towards the production of high-quality final components.
Author Response
Response to reviewers:
We would like to thank the reviewers for their excellent and fair reports and recommendations for improvement. All changes made to the manuscript are highlighted in yellow in the revised version of the manuscript and the comments are addressed each below:
Reviewer #3:
In paper Wire and Arc Additive Manufacturing (WAAM) is analise, various defects such as porosity, cracks, deformation and lack of fusion can occur during the fabrication process. These have a strong impact on the mechanical properties and can also lead to failure of the manufactured parts during service. These defects can be recognized using non-destructive testing (NDT) methods so that the examined workpiece is not harmed. This paper provides a comprehensive overview of various NDT techniques for WAAM, including laser-ultrasonic, acoustic emission with an airborne optical micro- phone, optical emission spectroscopy, laser-induced breakdown spectroscopy, laser-opto-ultrasonic dual detection, thermography and also in-process defect detection by weld current monitoring with an oscilloscope. In addition, the novel research conducted, its operating principle and the equipment required to perform these techniques are presented. The minimum defect size that can be identified by NDT methods has been obtained from previous academic research or from tests carried out by companies. The use of these techniques in WAAM applications makes it possible to detect defects and to take a step towards the production of high-quality final components.
The authors thank the reviewer for the careful consideration of the manuscript, and the recommendation for publishing the paper.
Masoud Shaloo
Junior Research Engineer | Center for Low-Emission Transport
LKR Leichtmetallkompetenzzentrum Ranshofen GmbH

Reviewer 4 Report
This manuscript submitted by Shaloo, et al., provides a comprehensive review of the NDT techniques and methods that have been applied for defect detection during the WAAM process. The structure and organization are generally clean and the references are rich. This work will be definitely helpful to the readers or engineers who are interested in the relevant field.
However, from the figures, the authors mainly focus on presenting the principle or experimental setup of each technique. No results were shown in the terms of figures. Therefore, the authors are strongly recommended to organize the remarkable result or representative output from the specific NDT study in one figure following the schematic illustration of each NDT method.
Moreover, please reconfirm the "suitable for online/offline monitoring" of each NDT method summed in Table 1. E.g., the acoustic emission (AE) is a typical passive method and is usually used as an online monitoring method. But by referring to line 114, how to realize offline monitoring using AE?
Author Response
Response to reviewers:
We would like to thank the reviewers for their excellent and fair reports and recommendations for improvement. All changes made to the manuscript are highlighted in yellow in the revised version of the manuscript and the comments are addressed each below:
Reviewer #4:
This manuscript submitted by Shaloo, et al., provides a comprehensive review of the NDT techniques and methods that have been applied for defect detection during the WAAM process. The structure and organization are generally clean and the references are rich. This work will be definitely helpful to the readers or engineers who are interested in the relevant field.
The authors thank the reviewer for the positive response.
However, from the figures, the authors mainly focus on presenting the principle or experimental setup of each technique. No results were shown in the terms of figures. Therefore, the authors are strongly recommended to organize the remarkable result or representative output from the specific NDT study in one figure following the schematic illustration of each NDT method.
At present, many of the described techniques have not been applied to the WAAM. Our future work will focus on testing and comparison of the results. This data will be subject of the future releases.
Moreover, please reconfirm the "suitable for online/offline monitoring" of each NDT method summed in Table 1. E.g., the acoustic emission (AE) is a typical passive method and is usually used as an online monitoring method. But by referring to line 114, how to realize offline monitoring using AE?
We have carefully checked table 1 for suitability for online/ offline monitoring and agree that regarding acoustic emission an error has been made, which has been corrected in the revised manuscript. An additional error was found and addressed in the current and voltage evaluation method.
Masoud Shaloo
Junior Research Engineer | Center for Low-Emission Transport
LKR Leichtmetallkompetenzzentrum Ranshofen GmbH

Round 2
Reviewer 2 Report
The authors have a minimal changes in the manuscript. Thye have changed the title of a review article without changing the content. If the original review was about wire arc additive manufacturing processes how can it changed to 'Application to fusion welding and future wire arc additive manufacturing processes' without changing the content.
There is no change in the article accept the titles.
Author Response
Response to reviewers (Round 2):
We would like to thank the reviewers for their excellent and fair reports and recommendations for improvement. All changes made to the manuscript are highlighted in green in the second revised version of the manuscript, and the comments are addressed each below:
Reviewer #2:
The authors have a minimal changes in the manuscript. They have changed the title of a review article without changing the content. If the original review was about wire arc additive manufacturing processes, how can it changed to 'Application to fusion welding and future wire arc additive manufacturing processes' without changing the content.
There is no change in the article accept the titles.
The authors would like to thank the reviewer for the careful review of the manuscript and useful comments. The authors modified the content of the review article to cover fusion welding processes as well. Several papers in the field of fusion welding have already been cited in the manuscript. Some examples are given below.
- Kah, B. Mvola, J. Martikainen, R. Suoranta, Real time non-destructive testing methods of welding, Adv. Mater. Res. 933 (2014) 571 109–116. https://doi.org/10.4028/www.scientific.net/AMR.933.109.
- Yusof, M.F. Jamaluddin, Welding Defects and Implications on Welded Assemblies, Compr. Mater. Process. 6 (2014) 125–134. 576 https://doi.org/10.1016/B978-0-08-096532-1.00605-1.
- Nomura, S. Otaki, R. Kita, S. Asai, In-situ detection of weld defect during the welding process by laser ultrasonic technique, Proc. Meet. Acoust. 38 (2019). https://doi.org/10.1121/2.0001171.
- Lévesque, Y. Asaumi, M. Lord, C. Bescond, H. Hatanaka, M. Tagami, J.-P. Monchalin, Inspection of thick welded joints using laser-ultrasonic SAFT, Ultrasonics. 69 (2016) 236–242. https://doi.org/10.1016/j.ultras.2016.04.001.
- G. Droubi, N.H. Faisal, F. Orr, J.A. Steel, M. El-Shaib, Acoustic emission method for defect detection and identification in carbon steel welded joints, J. Constr. Steel Res. 134 (2017) 28–37. https://doi.org/10.1016/j.jcsr.2017.03.012.
- Aboali, M. El-Shaib, A. Sharara, M. Shehadeh, Screening for welding defects using acoustic emission technique, Adv. Mater. Res. 1025–1026 (2014) 7–12. https://doi.org/10.4028/www.scientific.net/AMR.1025-1026.7.
However, to consider the reviewer's recommendations and enhance the quality of the paper, some papers are also included. Please find below some examples.
- Runnemalm, P. Broberg, E. Fernandez, A. Garcia, D.E.L.A. Yedra, P. Henrikson, N. Thorpe, Automatic thermography inspection of welded components with limited access, 6th Int. Symp. NDT Aerosp. (2014) 12–14.
- Sreedhar, C. V. Krishnamurthy, K. Balasubramaniam, V.D. Raghupathy, S. Ravisankar, Automatic defect identification using thermal image analysis for online weld quality monitoring, J. Mater. Process. Technol. 212 (2012) 1557–1566. https://doi.org/10.1016/j.jmatprotec.2012.03.002.
- Elkihel, A. Bakdid, F. Jeffali, H. Gziri, Evaluation of thermal losses for welded structures, Mater. Today Proc. 31 (2020) S78–S82. https://doi.org/10.1016/j.matpr.2020.06.098.
- Massaro, A. Panarese, G. Dipierro, E. Cannella, A. Galiano, Infrared Thermography and Image Processing applied on Weldings Quality Monitoring, 2020 IEEE Int. Work. Metrol. Ind. 4.0 IoT, MetroInd 4.0 IoT 2020 - Proc. (2020) 559–564. https://doi.org/10.1109/MetroInd4.0IoT48571.2020.9138310.
In the initial version of the manuscript, we cited various scientific works related to NDT of WAAM. Please find below some examples:
- Lopez, A., Bacelar, R., Pires, I., Santos, T.G., Sousa, J.P., Quintino, L.: Non-destructive testing application of 659 radiography and ultrasound for wire and arc additive manufacturing. Addit. Manuf. 21, 298–306 (2018). https://doi.org/10.1016/j.addma.2018.03.020
- Lukacs, P., Davis, G., Stratoudaki, T., Williams, S., MacLeod, C.N., Gachagan, A.: Remote Ultrasonic Imaging of 683 a Wire Arc Additive Manufactured Ti-6AI-4V Component using Laser Induced Phased Array. 1–6 (2021). https://doi.org/10.1109/i2mtc50364.2021.9459823
- Zeng, Y., Wang, X., Qin, X., Hua, L., Xu, M.: Laser Ultrasonic inspection of a Wire + Arc Additive Manufactured (WAAM) sample with artificial defects. Ultrasonics. 110, 106273 (2021)
- Zhang, C., Gao, M., Chen, C., Zeng, X.: Spectral diagnosis of wire arc additive manufacturing of Al alloys. Addit. Manuf. 30, 100869 (2019). https://doi.org/10.1016/j.addma.2019.100869
- Ma, Y., Hu, Z., Tang, Y., Ma, S., Chu, Y., Li, X., Luo, W., Guo, L., Zeng, X., Lu, Y.: Laser opto-ultrasonic dual 713 detection for simultaneous compositional, structural, and stress analyses for wire + arc additive manufacturing. Addit. Manuf. 31, 100956 (2020). https://doi.org/10.1016/j.addma.2019.100956
- Ma, Y., Hu, X., Hu, Z., Sheng, Z., Ma, S., Chu, Y., Wan, Q., Luo, W., Guo, L.: Simultaneous compositional and grain size measurements using laser opto-ultrasonic dual detection for additive manufacturing. Materials (Basel). 13, 1–10 (2020). https://doi.org/10.3390/ma13102404
However, in order to address the reviewer’s concerns following works are added.
- Seow, C.E., Zhang, J., Coules, H.E., Wu, G., Jones, C., Ding, J., Williams, S.: Effect of crack-like defects on the 706 fracture behaviour of Wire + Arc Additively Manufactured nickel-base Alloy 718. Manuf. 36, 101578 (2020). https://doi.org/10.1016/j.addma.2020.101578
- Wang, J., Cui, Y., Liu, C., Li, Z., Wu, Q., Fang, D.: Understanding internal defects in Mo fabricated by wire arc 709 additive manufacturing through 3D computed tomography. Alloys Compd. 840, 155753 (2020). https://doi.org/10.1016/j.jallcom.2020.155753
- Bento, J.B., Lopez, A., Pires, I., Quintino, L., Santos, T.G.: Non-destructive testing for wire + arc additive 712 manufacturing of aluminium parts. Addit. Manuf. 29, 100782 (2019). https://doi.org/10.1016/j.addma.2019.100782
- Lopez, A.B., Santos, J., Sousa, J.P., Santos, T.G., Quintino, L.: Phased Array Ultrasonic Inspection of Metal Additive Manufacturing Parts. J. Nondestruct. Eval. 38, 1–11 (2019). https://doi.org/10.1007/s10921-019-0600-y
- Chabot, A., Laroche, N., Carcreff, E., Rauch, M., Hascoët, J.Y.: Towards defect monitoring for metallic additive manufacturing components using phased array ultrasonic testing. J. Intell. Manuf. 31, 1191–1201 (2020). 718 https://doi.org/10.1007/s10845-019-01505-9
- Javadi, Y., Macleod, C.N., Pierce, S.G., Gachagan, A., Lines, D., Mineo, C., Ding, J., Williams, S., Vasilev, M., Mohseni, E., Su, R.: Ultrasonic phased array inspection of a Wire + Arc Additive Manufactured ( WAAM ) sample with intentionally embedded defects. Addit. Manuf. 29, 100806 (2019). https://doi.org/10.1016/j.addma.2019.100806
- Ramalho, A., Santos, T.G., Bevans, B., Smoqi, Z., Rao, P., Oliveira, J.P.: Effect of contaminations on the acoustic emissions during wire and arc additive manufacturing of 316L stainless steel. Addit. Manuf. 51, (2022). https://doi.org/10.1016/j.addma.2021.102585
- Bacelar, R.: Evaluation of WAAM parts by NDT (MSc Thesis). (2017)
- Yang, D., Wang, G., Zhang, G.: Thermal analysis for single-pass multi-layer GMAW based additive manufacturing using infrared thermography. J. Mater. Process. Technol. 244, 215–224 (2017). https://doi.org/10.1016/j.jmatprotec.2017.01.024
Masoud Shaloo
Junior Research Engineer | Center for Low-Emission Transport
LKR Leichtmetallkompetenzzentrum Ranshofen GmbH

Reviewer 4 Report
Authors' response:
At present, many of the described techniques have not been applied to the WAAM. Our future work will focus on testing and comparison of the results. This data will be subject of the future releases.
Additional comments:
Apparently, the reason "have not been applied" is not acceptable.
Please see some literature data below that were obtained from a random googling.
Liquid penetrant testing, ultrasonic testing, radiography:
Lopez, Ana, et al. "Non-destructive testing application of radiography and ultrasound for wire and arc additive manufacturing." Additive Manufacturing 21 (2018): 298-306.
Acoustic emission testing:
Ramalho, André, et al. "Effect of contaminations on the acoustic emissions during wire and arc additive manufacturing of 316L stainless steel." Additive Manufacturing 51 (2022): 102585.
Ito, Kaita, et al. "Detection and location of microdefects during selective laser melting by wireless acoustic emission measurement." Additive Manufacturing 40 (2021): 101915.
Laser ultrasonic testing
Ma, Yuyang, et al. "Laser opto-ultrasonic dual detection for simultaneous compositional, structural, and stress analyses for wire+ arc additive manufacturing." Additive Manufacturing 31 (2020): 100956.
Phased array ultrasonic testing
Lopez, Ana Beatriz, et al. "Phased array ultrasonic inspection of metal additive manufacturing parts." Journal of Nondestructive Evaluation 38.3 (2019): 1-11.
Chabot, Alexia, et al. "Towards defect monitoring for metallic additive manufacturing components using phased array ultrasonic testing." Journal of Intelligent Manufacturing 31.5 (2020): 1191-1201.
Computed tomography
Wang, Jiachen, et al. "Understanding internal defects in Mo fabricated by wire arc additive manufacturing through 3D computed tomography." Journal of Alloys and Compounds 840 (2020): 155753.
Infrared thermography
Shen, Chen, et al. "Composition-induced microcrack defect formation in the twin-wire plasma arc additive manufacturing of binary TiAl alloy: An X-ray computed tomography-based investigation." Journal of Materials Research (2021): 1-12.
Infrared Thermography
Yang, Dongqing, Gang Wang, and Guangjun Zhang. "Thermal analysis for single-pass multi-layer GMAW based additive manufacturing using infrared thermography." Journal of Materials Processing Technology 244 (2017): 215-224.
Optical emission spectroscopy
Mahadevan, Gautham. "Optical Emission Spectroscopy during Wire and Arc Additive Manufacturing." (2020).
As a review work, introducing the principle or experimental setup of NDT techniques only is not good enough for publication. Some representative results of each NDT method must be included to demonstrate their feasibility, advancements, differences, as well as challenges.
By the way, the literature above listed here does not mean that the authors must cite them; instead, it is just to prove that the NDT techniques reviewed have been applied to WAAM. The authors should improve the manuscript according to this point.
Author Response
Response to reviewers (Round 2):
We would like to thank the reviewers for their excellent and fair reports and recommendations for improvement. All changes made to the manuscript are highlighted in green in the second revised version of the manuscript, and the comments are addressed each below:
Reviewer #4:
Authors' response:
At present, many of the described techniques have not been applied to the WAAM. Our future work will focus on testing and comparison of the results. This data will be subject of the future releases.
Additional comments:
Apparently, the reason "have not been applied" is not acceptable.
Please see some literature data below that were obtained from a random googling.
Liquid penetrant testing, ultrasonic testing, radiography:
Lopez, Ana, et al. "Non-destructive testing application of radiography and ultrasound for wire and arc additive manufacturing." Additive Manufacturing 21 (2018): 298-306.
….
As a review work, introducing the principle or experimental setup of NDT techniques only is not good enough for publication. Some representative results of each NDT method must be included to demonstrate their feasibility, advancements, differences, as well as challenges.
By the way, the literature above listed here does not mean that the authors must cite them; instead, it is just to prove that the NDT techniques reviewed have been applied to WAAM. The authors should improve the manuscript according to this point.
The authors would like to thank the reviewer for the useful comments. In order to meet reviewer’s concerns, we added the following figures:
- Figure 1. A combination of radiographic and image processing techniques for welding defect detection conducted by Faramarzi et al.
- Figure 2. A comparison between dye penetrant testing (A), conventional ultrasonic (B) and x-ray radiographic technologies (C) carried out by Seow et al.
- Figure 3. Inspected WAAM Component
- Figure 4. Defect detection by means of TFM image of the component using ultrasonic longitudinal waves
- Figure 9. A correlation between acoustic emission absolute energy and welding defect
- Figure 10. A schematic of the OES technique
- Figure 15. Defect detection by means of the thermography technology performed by Broberg
- Figure 20. A comparison between one data block of normal and abnormal arc voltage in GTAW conducted by Zhang et al.
In the initial version of the manuscript, we have cited various scientific works related to NDT of WAAM. Please find below some examples:
- Lopez, A., Bacelar, R., Pires, I., Santos, T.G., Sousa, J.P., Quintino, L.: Non-destructive testing application of radiography and ultrasound for wire and arc additive manufacturing. Addit. Manuf. 21, 298–306 (2018). https://doi.org/10.1016/j.addma.2018.03.020
- Lukacs, P., Davis, G., Stratoudaki, T., Williams, S., MacLeod, C.N., Gachagan, A.: Remote Ultrasonic Imaging of a Wire Arc Additive Manufactured Ti-6AI-4V Component using Laser Induced Phased Array. 1–6 (2021). https://doi.org/10.1109/i2mtc50364.2021.9459823
- Zeng, Y., Wang, X., Qin, X., Hua, L., Xu, M.: Laser Ultrasonic inspection of a Wire + Arc Additive Manufactured (WAAM) sample with artificial defects. Ultrasonics. 110, 106273 (2021)
- Zhang, C., Gao, M., Chen, C., Zeng, X.: Spectral diagnosis of wire arc additive manufacturing of Al alloys. Addit. Manuf. 30, 100869 (2019). https://doi.org/10.1016/j.addma.2019.100869
- Ma, Y., Hu, Z., Tang, Y., Ma, S., Chu, Y., Li, X., Luo, W., Guo, L., Zeng, X., Lu, Y.: Laser opto-ultrasonic dual detection for simultaneous compositional, structural, and stress analyses for wire + arc additive manufacturing. Addit. Manuf. 31, 100956 (2020). https://doi.org/10.1016/j.addma.2019.100956
- Ma, Y., Hu, X., Hu, Z., Sheng, Z., Ma, S., Chu, Y., Wan, Q., Luo, W., Guo, L.: Simultaneous compositional and grain size measurements using laser opto-ultrasonic dual detection for additive manufacturing. Materials (Basel). 13, 1–10 (2020). https://doi.org/10.3390/ma13102404
However, in order to address the reviewer’s concerns some suggested papers by the reviewer, which are related to defect detection in WAAM, are also included.
- Wang, J., Cui, Y., Liu, C., Li, Z., Wu, Q., Fang, D.: Understanding internal defects in Mo fabricated by wire arc 709 additive manufacturing through 3D computed tomography. J. Alloys Compd. 840, 155753 (2020). https://doi.org/10.1016/j.jallcom.2020.155753
- Bento, J.B., Lopez, A., Pires, I., Quintino, L., Santos, T.G.: Non-destructive testing for wire + arc additive 712 manufacturing of aluminium parts. Addit. Manuf. 29, 100782 (2019). https://doi.org/10.1016/j.addma.2019.100782
- Lopez, A.B., Santos, J., Sousa, J.P., Santos, T.G., Quintino, L.: Phased Array Ultrasonic Inspection of Metal Additive Manufacturing Parts. J. Nondestruct. Eval. 38, 1–11 (2019). https://doi.org/10.1007/s10921-019-0600-y
- Chabot, A., Laroche, N., Carcreff, E., Rauch, M., Hascoët, J.Y.: Towards defect monitoring for metallic additive manufacturing components using phased array ultrasonic testing. J. Intell. Manuf. 31, 1191–1201 (2020). 718 https://doi.org/10.1007/s10845-019-01505-9
- Javadi, Y., Macleod, C.N., Pierce, S.G., Gachagan, A., Lines, D., Mineo, C., Ding, J., Williams, S., Vasilev, M., Mohseni, E., Su, R.: Ultrasonic phased array inspection of a Wire + Arc Additive Manufactured ( WAAM ) sample with intentionally embedded defects. Addit. Manuf. 29, 100806 (2019). https://doi.org/10.1016/j.addma.2019.100806
- Ramalho, A., Santos, T.G., Bevans, B., Smoqi, Z., Rao, P., Oliveira, J.P.: Effect of contaminations on the acoustic emissions during wire and arc additive manufacturing of 316L stainless steel. Addit. Manuf. 51, (2022). https://doi.org/10.1016/j.addma.2021.102585
- Seow, C.E., Zhang, J., Coules, H.E., Wu, G., Jones, C., Ding, J., Williams, S.: Effect of crack-like defects on the 706 fracture behaviour of Wire + Arc Additively Manufactured nickel-base Alloy 718. Addit. Manuf. 36, 101578 (2020). https://doi.org/10.1016/j.addma.2020.101578
- Bacelar, R.: Evaluation of WAAM parts by NDT (MSc Thesis). (2017)
- Yang, D., Wang, G., Zhang, G.: Thermal analysis for single-pass multi-layer GMAW based additive manufacturing using infrared thermography. J. Mater. Process. Technol. 244, 215–224 (2017). https://doi.org/10.1016/j.jmatprotec.2017.01.024
In addition, the novel conducted research in the area of the thermography technology has been also added. Please find below some examples:
- Runnemalm, P. Broberg, E. Fernandez, A. Garcia, D.E.L.A. Yedra, P. Henrikson, N. Thorpe, Automatic thermography inspection of welded components with limited access, 6th Int. Symp. NDT Aerosp. (2014) 12–14.
- U. Sreedhar, C. V. Krishnamurthy, K. Balasubramaniam, V.D. Raghupathy, S. Ravisankar, Automatic defect identification using thermal image analysis for online weld quality monitoring, J. Mater. Process. Technol. 212 (2012) 1557–1566. https://doi.org/10.1016/j.jmatprotec.2012.03.002.
- A. Elkihel, A. Bakdid, F. Jeffali, H. Gziri, Evaluation of thermal losses for welded structures, Mater. Today Proc. 31 (2020) S78–S82. https://doi.org/10.1016/j.matpr.2020.06.098.
- A. Massaro, A. Panarese, G. Dipierro, E. Cannella, A. Galiano, Infrared Thermography and Image Processing applied on Weldings Quality Monitoring, 2020 IEEE Int. Work. Metrol. Ind. 4.0 IoT, MetroInd 4.0 IoT 2020 - Proc. (2020) 559–564. https://doi.org/10.1109/MetroInd4.0IoT48571.2020.9138310.
Masoud Shaloo
Junior Research Engineer | Center for Low-Emission Transport
LKR Leichtmetallkompetenzzentrum Ranshofen GmbH

Round 3
Reviewer 2 Report
The revised article has improved in the latest version. The Conclusions section still needs improvement. The technique-specific observations, merits, demerits, and possibilities should be listed pointwise to make the reader understand the article's central theme. The Abstract and Conclusions in combination should enable the reader to go or not go for the full reading.
The article may be accepted after the changes in the Conclusions section, as suggested.
The revised article has improved in the latest version. The Conclusions section still needs improvement. The technique-specific observations, merits, demerits, and possibilities should be listed pointwise to make the reader understand the article's central theme. The Abstract and Conclusions in combination should enable the reader to go or not go for the full reading.
Author Response
Response to reviewers (Round 3):
We would like to thank the reviewers for their excellent and fair reports and recommendations for improvement. All changes made to the manuscript are highlighted in red in the third revised version of the manuscript, and the comments are addressed each below:
Reviewer #2:
The revised article has improved in the latest version. The Conclusions section still needs improvement. The technique-specific observations, merits, demerits, and possibilities should be listed pointwise to make the reader understand the article's central theme. The Abstract and Conclusions in combination should enable the reader to go or not go for the full reading.
The article may be accepted after the changes in the Conclusions section, as suggested.
We would like to thank the reviewer for the careful review of the manuscript and useful comments. We modified the conclusion of the review article to meet the reviewers’s concerns.
Masoud Shaloo
Junior Research Engineer | Center for Low-Emission Transport
LKR Leichtmetallkompetenzzentrum Ranshofen GmbH

Reviewer 4 Report
Dear Authors,
Thank you for your efforts to improve the manuscript. All my concern has been cleared and I would like to recommend it be published according to the revised version.
Regards.
Author Response
Response to reviewers (Round 3):
Reviewer #4:
Dear Authors,
Thank you for your efforts to improve the manuscript. All my concern has been cleared and I would like to recommend it be published according to the revised version.
Regards.
The authors thank the reviewer for the careful consideration of the manuscript, and the recommendation for publishing the paper.
Masoud Shaloo
Junior Research Engineer | Center for Low-Emission Transport
LKR Leichtmetallkompetenzzentrum Ranshofen GmbH
